# PROMPTING GPT-3 TO BE RELIABLE

**Chenglei Si**[1]*, **Zhe Gan**[2], **Zhengyuan Yang**[2], **Shuohang Wang**[2]
**Jianfeng Wang**[2], **Jordan Boyd-Graber**[1], **Lijuan Wang**[2]
[1] University of Maryland      [2] Microsoft
  clsi@umd.edu  pkuganzhe@gmail.com  lijuanw@microsoft.com

## ABSTRACT

Large language models (LLMs) show impressive abilities via few-shot prompting. Commercialized APIs such as OpenAI GPT-3 further increase their use in real-world language applications. However, the crucial problem of how to improve the *reliability* of GPT-3 is still under-explored. While reliability is a broad and vaguely defined term, we decompose reliability into four main facets that correspond to the existing framework of ML safety and are well-recognized to be important: generalizability, social biases, calibration, and factuality. Our core contribution is to establish simple and effective prompts that improve GPT-3's reliability as it: 1) generalizes out-of-distribution, 2) balances demographic distribution and uses natural language instructions to reduce social biases, 3) calibrates output probabilities, and 4) updates the LLM's factual knowledge and reasoning chains. With appropriate prompts, GPT-3 is more reliable than smaller-scale supervised models on all these facets. We release all processed datasets, evaluation scripts, and model predictions.[1] Our systematic empirical study not only sheds new insights on the reliability of prompting LLMs, but more importantly, our prompting strategies can help practitioners more reliably use LLMs like GPT-3.

## 1 INTRODUCTION

NLP is dominated by large language models (LLMs) — pretrained on large, unlabeled text data — that are then used for downstream tasks (Devlin et al., 2019a; Brown et al., 2020). Scaling the model and data size often brings gains on downstream tasks (Kaplan et al., 2020; BIG-Bench, 2022), allowing what some call emergent abilities (Wei et al., 2022a). These emergent behaviors are accomplished through prompting—a crafted, natural language text to shape predictions or offer relevant information without expensive supervised data. Among all the existing LLMs, GPT-3 (Brown et al., 2020) is particularly popular due to its flexibility and ease of use from the OpenAI API [2].

Existing empirical studies investigate GPT-3 on specific tasks such as mathematical reasoning (Hendrycks et al., 2021a), multi-hop reasoning (Wei et al., 2022b; Kojima et al., 2022), and code generation (Chen et al., 2021a). However, rising numbers on these evaluations do not ensure LLM *reliability*. For example, LLMs (including GPT-3) produce biased (Lucy & Bamman, 2021) generations, false statements (Lin et al., 2022b), and outdated information (Chen et al., 2021b; Kasai et al., 2022). Deploying such models in the real world could result in catastrophic harm.

In the context of prompting LLMs, several previous works have explored their reliability. For example, in the release reports of GPT-3 (Brown et al., 2020), OPT (Zhang et al., 2022), Gopher (Rae et al., 2021) and PaLM (Chowdhery et al., 2022), there are dedicated experiments evaluating these LLMs' representational bias and toxicity. Another line of work has evaluated calibration (Lin et al., 2022a; Kadavath et al., 2022) of prompting-based LLMs on math questions or multiple-choice questions. We differ from these prior works in two key aspects: ($i$) We perform a more comprehensive study of four core facets of reliability, serving as a meta-analysis. ($ii$) We focus particularly on find-

---

*Work done during internship at Microsoft.

[1] https://github.com/NoviScl/GPT3-Reliability

[2] By default, we use the CODE-DAVINCI-002 model (also known as Codex or GPT 3.5) in our experiments unless otherwise specified, because our preliminary results show that this is the most accurate model on most NLP datasets we tried.

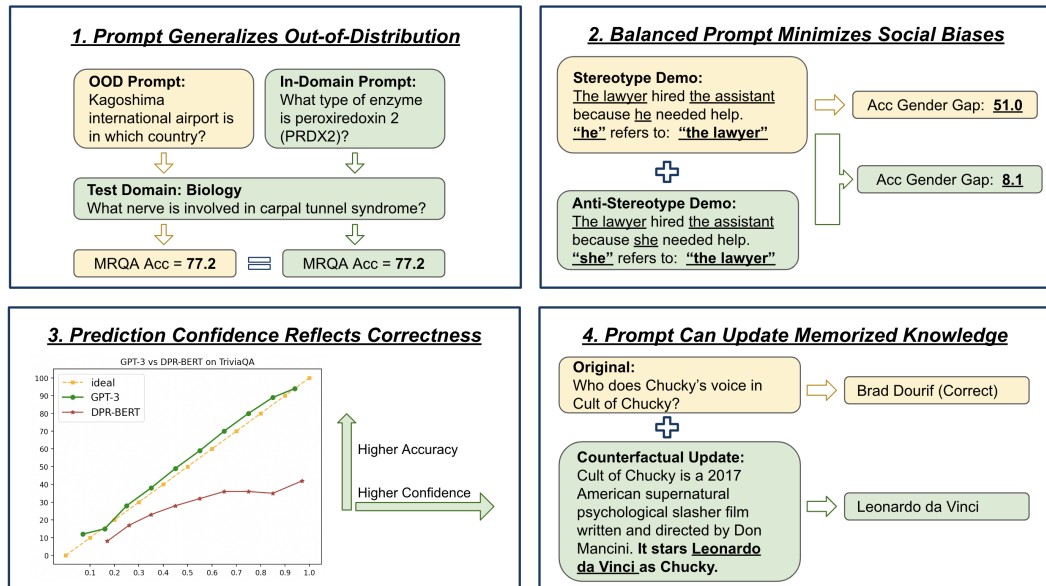

Figure 1: Four main reliability factors we examined and the core findings.

ing prompting strategies that are effective under these reliability facets, rather than just evaluating intrinsic model characteristics (Figure 1).

Our reliability testing framework takes inspiration from the survey of unsolved problems in ML safety (Hendrycks et al., 2021b): withstanding hazards (generalizability), identifying hazards (calibration), steering ML systems and reducing deployment hazards (reducing social biases and improving factuality). These facets also aim to address the risks of ML systems identified in existing conceptual frameworks (Tan et al., 2022; 2021). **We have a more extensive discussion of related works in Appendix Section A.**

As summarized in Figure 1, our simple prompting strategies beat smaller-scale supervised models on all reliability metrics we consider: 1) prompting with randomly sampled examples from the source domain allows GPT-3 to generalize robustly on unseen domains and challenge examples; 2) examples sampled from a balanced demographic distribution and natural language intervention reduce social biases; 3) language model probabilities are calibrated to reflect accuracy; and 4) appending up-to-date knowledge can supplant GPT-3's memorized knowledge or reasoning chains.

## 2 FACET 1: GENERALIZABILITY

LLMs are often criticized for missing the forest for the trees. They overfit training data from a particular domain (domain shift), are not robust to minor changes in a text (perturbations), or use shortcuts to make predictions (spurious correlations). These pathologies make models unreliable since these distribution shifts happen all the time in real-world data and could incur significant performance drops. In this section, we study whether GPT-3 can stay robust when the test data come from different distributions than the demo examples in the prompt, and how their generalization compares to supervised models.

**Experiment Setup**  We study all three types of distribution shifts mentioned above. For each of them, researchers have created datasets that target modern language models' weaknesses which we adopt for evaluation. For **domain shift**, MRQA (Fisch et al., 2019) trains on six machine reading datasets from the source domain and tests on six different target domains; for **perturbations**, AdvGLUE (Wang et al., 2021) craft adversarial versions of GLUE (Wang et al., 2018) based on automatic adversarial perturbations and human filtering, and Contrast Sets (Gardner et al., 2020) are expert-authored minimal edits that change the label; for **spurious correlation**, HANS (McCoy et al., 2019) and PAWS (Zhang et al., 2019) are challenge sets designed for models trained on MNLI and

| | MRQA | | | AdvGLUE | | | Contrast Set | | |
|---|---|---|---|---|---|---|---|---|---|
| | Source↑ | Target↑ | Gap↓ | Original↑ | Perturbed↑ | Gap↓ | Original↑ | Perturbed↑ | Gap↓ |
| RoBERTa | 81.6 | 62.1 | 19.5 | 91.7 | 51.7 | 40.0 | 86.1 | 71.1 | 15.0 |
| GPT-3 | 79.8 | 77.2 (S) / 77.2 (T) | **2.6** | 84.2 | 69.3 | **14.9** | 85.5 | 80.0 | **5.5** |

Table 1: For GPT-3 on MRQA target domain test sets, we report results for using both demos from the source (S) and target (T) domains, which surprisingly achieve the same F1 on the target domains (77.2). For AdvGLUE and Contrast Set, we use accuracy as the metric and we use demos from the clean data as the prompt. In all cases, GPT-3 few-shot prompting incurs much smaller performance gaps on the OOD or challenge test sets than the supervised RoBERTa (123M) baseline. We include the comparison with many other supervised baselines in Appendix B and GPT-3 exhibits better generalization than all of them.

QQP where the lexical overlap feature in the training data does not hold during testing. For each of these settings, we evaluate a simple prompting strategy by sampling examples from the source domains (for MRQA, we use a fixed prompt consisting of eight randomly sampled examples from the source domain on all target datasets; for perturbations and spurious correlation, we randomly sample 16 demos from the original clean training data from GLUE, MNLI, and QQP respectively). In addition, for domain shift, we also consider a prompt where we sample eight examples from the training set of each target domain to ablate the impact of the distribution of the demo examples.

**Results** Table 1 and Table 2 compare supervised RoBERTa (Liu et al., 2019) and BERT (Devlin et al., 2019b) models trained on the source domain datasets or the clean training data with GPT-3 that uses examples sampled from the same training data as in the supervised models. [3] GPT-3 achieves higher accuracy on the OOD tests even when it is slightly worse on the in-domain test sets than the supervised baselines, leading to smaller generalization gaps. This shows that prompting GPT-3 can be more robust than supervised finetuning of smaller-scale language models. Surprisingly, we compare using demo examples sampled from the source domains versus target domains on MRQA, and both prompting methods give the same OOD generalization results, indicating that GPT-3 prompts can directly generalize to OOD test sets where the test examples are from a different distribution than the prompt demo distribution, possibly because the role of demonstration examples is more in specifying the task rather than informing the input distribution (Min et al., 2022).

| | BERT (340M) | RoBERTa (354M) | GPT-3 |
|---|---|---|---|
| | *MNLI → HANS* | | |
| MNLI↑ | 86.2 | 89.1 | 77.6 |
| HANS↑ | 71.4 | 77.1 | 75.3 |
| Gap↓ | 14.8 | 12.0 | **2.3** |
| | *QQP → PAWS* | | |
| QQP↑ | 91.3 | 89.0 | 83.5 |
| PAWS↑ | 40.1 | 39.5 | 73.7 |
| Gap↓ | 51.2 | 49.5 | **9.8** |

Table 2: When using demos sampled from MNLI and QQP, GPT-3 few-shot prompting achieves much better generalization than smaller supervised models (BERT and RoBERTa) on the OOD test sets HANS and PAWS.

**Takeaway** (*i*) Few-shot prompting of GPT-3 is more robust than supervised models such as finetuned BERT and RoBERTa, under all three settings (domain shift, perturbations, spurious correlation). (*ii*) Using randomly sampled demos from the source datasets is a simple but strong baseline, in fact, it performs the same as using demos sampled from the target distributions.

## 3 FACET 2: SOCIAL BIAS AND FAIRNESS

Apart from high performance on in-domain and OOD datasets, the second key facet of reliability is that we expect models to be fair to different demographic groups. Biased models cause severe harm when deployed in real-world applications, especially to the minority groups being discriminated against (Cao et al., 2022). In this section, we examine whether GPT-3 produces biased predictions in two downstream tasks - coreference resolution and question answering.

---

[3]For Contrast Set, we show results on the BoolQ subset in the main paper and present results for other subsets in Table 14, which gives the same conclusion that GPT-3 is more robust than supervised models.

| Prompt | Type I Pro$_\uparrow$ | Type I Anti$_\uparrow$ | Gap$_{|\downarrow|}$ | Type II Pro$_\uparrow$ | Type II Anti$_\uparrow$ | Gap$_{|\downarrow|}$ |
|---|---|---|---|---|---|---|
| *Supervised Baseline* | | | | | | |
| E2E (Lee et al., 2017) | 74.9 | 47.4 | 27.2 | 88.6 | 77.3 | 11.3 |
| GPT-3 Few-Shot: *Bias Distribution in the Prompt (16 shots)* | | | | | | |
| Balanced | 89.2 | 81.1 | 8.1 | 99.2 | 95.5 | 3.7 |
| Type I - Pro | 93.4 | 42.4 | 51.0 | 91.1 | 78.9 | 12.2 |
| Type II - Pro | 87.6 | 59.5 | 28.1 | 100.0 | 98.7 | 1.3 |
| Type I - Anti | 50.8 | 80.8 | -30.0 | 57.4 | 51.1 | 6.3 |
| Type II - Anti | 85.5 | 68.2 | 17.3 | 100.0 | 99.5 | 0.5 |
| GPT-3 Few-Shot: *Prompt Ordering (16 shots, Balanced)* | | | | | | |
| Randomly Shuffled | 89.2 | 81.1 | 8.1 | 99.2 | 95.5 | 3.7 |
| Pro in the end | 89.5 | 76.3 | 13.2 | 93.7 | 81.8 | 11.9 |
| Anti in the end | 94.2 | 73.2 | 21.0 | 95.5 | 87.1 | 8.4 |

Table 3: GPT-3 results on WinoBias. The bias gap between Pro-Bias subsets and Anti-Bias subsets indicates the extent of gender biases exhibited by the model (smaller-scale gap is better). Results of the baseline ECE model (Lee et al., 2017) are taken from the WinoBias paper (Zhao et al., 2018). Prompt with balanced pro-bias and anti-bias answers best shrinks the bias gap, and randomly shuffling the demos is better than putting one group at the end.

## 3.1 THE CASE OF GENDER BIAS: WINOBIAS

**Dataset** We start with the WinoBias dataset (Zhao et al., 2018) which uses templates to check whether models are more likely to assign gender pronouns to stereotypical occupations. WinoBias has two types of examples: Type I are ambiguous, challenging examples that require world knowledge; Type II can be resolved using only syntactic information. For each type, examples either confirm (pro-bias) or challenge (anti-bias) societal biases. Ideally, coreference accuracy should be similar on the pro-bias and anti-bias subsets (small gaps).

**Prompt Design** For ease of evaluation, we re-format the WinoBias dataset into a question-answering format where we provide the original sentence and then add a question *"What does the pronoun refer to in the above sentence?"* (*"the pronoun"* is replaced with the actual pronoun in the sentence) and we use the answer exact match as the evaluation metric. We randomly sample examples from the training set as the prompt and then evaluate on the Pro and Anti test sets.

**Which Examples Should be in the Prompt** We compare: 1) sampling four demo examples from each of the Type I-Pro, Type I-Anti, Type II-Pro, and Type II-Anti subsets (Balanced), which results in a total of 16 demos; 2) sampling 16 demo examples from a single subset. The balanced prompt induces the least biased predictions (Table 3, second block). In particular, if we only keep Pro-Bias examples, the model will favor Pro-Bias predictions (especially on Type I test examples because they are more ambiguous while Type II examples have clear syntax cues).

**How Should Examples be Ordered** We compare: 1) randomly shuffling the demo examples; and 2) putting all Pro-Bias or Anti-Bias examples at the end of the prompt. Random shuffling reduces bias gaps most (Table 3, third block). Interestingly, putting either Pro-Bias or Anti-Bias examples at the end increases bias gaps.

## 3.2 BROADER SOCIAL DIMENSIONS: BBQ

**Dataset** We now explore additional social dimensions using BBQ (Parrish et al., 2022), which tests social biases against people from nine protected classes (age, disability status, gender identity, nationality, physical appearance, race, religion, socio-economic status, sexual orientation). BBQ examples are in sets of four multiple-choice questions. Two questions are ambiguous—the context lacks evidence to point to an answer. Two other questions in each set have a context that points to an unambiguous answer: the model should choose the correct answer rather than abstaining. Each question has three options: a pro-bias answer that supports the stereotype, an anti-bias answer

| Prompt | Ambig Acc$_\uparrow$ | DisAmbig Acc$_\uparrow$ | Ambig Bias Score$_{|\downarrow|}$ | DisAmbig Bias Score$_{|\downarrow|}$ |
|---|---|---|---|---|
| *Supervised Baselines* | | | | |
| RoBERTa-Base (123M) | 61.2 | 52.7 | 4.9 | 4.7 |
| RoBERTa-Large (354M) | 49.4 | 87.3 | 10.4 | 1.2 |
| DeBERTa-Base (184M) | 47.6 | 90.4 | 12.8 | 2.9 |
| DeBERTa-Large (435M) | 30.1 | 95.5 | 24.7 | -1.0 |
| *GPT-3 Few-Shot Prompting* | | | | |
| 0-shot | 60.5 | 43.2 | 3.7 | 4.4 |
| BBQ Balanced | 96.8 | 76.0 | 2.4 | 1.5 |
| BBQ Ambig-Neutral | 99.9 | 13.2 | 0.0 | -3.5 |
| BBQ Ambig-Pro-Bias | 2.6 | 97.3 | 24.7 | 3.2 |
| BBQ Ambig-Anti-Bias | 2.5 | 97.0 | 23.6 | 3.1 |

Table 4: Results on the BBQ dataset. For GPT-3 prompting, apart from the zero-shot result, others use 8-shots. For accuracy (Acc), higher value is better; for bias score, lower magnitude is better. The balanced prompt best trades-off accuracy and bias for GPT-3.

| Prompt | Ambig Acc$_\uparrow$ | DisAmbig Acc$_\uparrow$ | Ambig Bias Score$_\downarrow$ | DisAmbig Bias Score$_\downarrow$ |
|---|---|---|---|---|
| Before Intervention | 2.6 | 97.3 | 24.7 | 3.2 |
| After Intervention | 96.6 | 51.5 | 1.9 | 3.8 |

Table 5: The impact of adding natural language intervention to a biased prompt on GPT-3. Adding an instruction leads the model to make more neutral predictions and reduce bias scores.

that counters the stereotype, and a neutral answer (*e.g.*, "*Cannot be determined.*"). When reporting results, we report: 1) accuracy on ambiguous and unambiguous questions (higher is better); 2) bias scores on ambiguous and disambiguated questions (smaller scale is better). Intuitively, the bias score measures the frequency of the model predicting a pro-bias answer when it makes a non-unknown prediction, where 0% means no bias and 100% means always following the bias.

**Answer Distribution Affects Biases** We sample the same number of demo examples with different answer distributions (Table 4). With zero-shot inference, GPT-3's low bias is balanced by low accuracy. A balanced prompt—where we sample both ambiguous and disambiguated questions with pro-bias and anti-bias answers—increases accuracy while maintaining low bias scores. Next, when we only present ambiguous questions with neutral answers in the prompt, GPT-3 can easily pick up this pattern to always predict the neutral answer, resulting in extremely low bias scores and high accuracy on ambiguous questions, but sacrificing accuracy on the disambiguated questions. Lastly, we experiment with using biased prompts where we only sample ambiguous questions in the prompt, and we set their answers to either all bias-supporting or bias-countering. As shown in the last two rows in Table 4, both cases lead to a biased model that gets high bias scores on ambiguous questions. This shows that when the demo examples are biased towards certain demographic groups, GPT-3 will pick up such biases, and thus one should prefer a balanced prompt in downstream applications.

**Natural Language Intervention Mitigates Biases** Lastly, we explore the possibility of using natural language instructions to guide the predictions of GPT-3 towards better fairness. We use the Ambig-Pro-Bias prompt and prepend the additional instruction: "*We should treat people from different socioeconomic statuses, sexual orientations, religions, races, physical appearances, nationalities, gender identities, disabilities, and ages equally. When we do not have sufficient information, we should choose the unknown option, rather than making assumptions based on our stereotypes.*" to every test question. In Table 5, after adding this intervention, GPT-3 prefers the neutral answer much more often, leading to a much higher accuracy on the ambiguous questions, and at the same time significantly reducing the bias scores. This shows that GPT-3 is sensitive to such natural language intervention. This is in contrast with smaller language models such as RoBERTa (Zhao et al., 2021a), which is more rigid. This finding offers a new way for effectively reducing social biases.

**Takeaway** (*i*) Demographic distribution of answers has huge impact on models' biases, sampling balanced prompt best reduces biases. (*ii*) Randomly shuffling the demos leads to smaller biases than putting all pro-bias or anti-bias examples in the end. (*iii*) Specifying intended model behaviors such as being fair via instructions in the prompt can effectively guide model predictions.

## 4 FACET 3: UNCERTAINTY CALIBRATION

No language model can ever be perfect, and to safely use these imperfect models, users must decide when to **trust** model predictions to avoid mistrusting wrong predictions, especially in high-stake settings. This requires another facet of reliability - uncertainty calibration: providing confidence scores for each model prediction that accurately reflects the likelihood of the predicted answer being correct.

### 4.1 EVALUATION SETUP

**Experiment Setup** We study the setting of free-form answer generation: given a test question, we prompt the model to generate an answer string and obtain its confidence score (more below), and we evaluate the correctness of the generated answer based on exact match with the gold answer. We experiment with three QA datasets: NQ, TriviaQA, and HotpotQA. In all cases, we adopt the closed-book setting (i.e., no additional evidence passages). We focus on intrinsic calibration results: using raw confidence scores rather than post-hoc calibration, which requires an additional dev set for parameter-tuning. We report the standard calibration metric expected calibration error (ECE), the reliability diagram,[4] and selective prediction results where we rank all predictions by their confidence and see if the accuracy of the most confident predictions is significantly higher than the average accuracy. Because of ECE's known flaws due to its bucketing mechanism (Si et al., 2022), so we also report the Brier score (Brier, 1950). Our baseline is a supervised QA model—DPR-BERT (Si et al., 2022)—with a dense passage retriever (DPR; Karpukhin et al., 2020) to feed the top passages into a BERT reader model for answer extraction. We follow their joint calibration setup for scoring predictions of DPR-BERT.

|  | Acc$_\uparrow$ | ECE$_\downarrow$ | Brier$_\downarrow$ |
|---|---|---|---|
| **NQ** | | | |
| DPR-BERT (110M) | 36.1 | 29.4 | 33.5 |
| GPT-3 LM Prob | 40.5 | 18.9 | 23.3 |
| GPT-3 Self-Con | 40.2 | 14.3 | 20.1 |
| **TriviaQA (TQA)** | | | |
| GPT-3 LM Prob | 73.8 | 3.8 | 15.9 |
| GPT-3 Self-Con | 73.2 | 11.9 | 16.5 |
| **HotpotQA (HQA)** | | | |
| GPT-3 LM Prob | 29.8 | 25.0 | 23.5 |
| GPT-3 Self-Con | 28.5 | 20.7 | 19.9 |
| **Different Prompts on NQ w/ LM-Prob** | | | |
| GPT-3 2-shot | 37.0 | 11.7 | 20.8 |
| GPT-3 4-shot | 38.3 | 13.4 | 21.0 |
| GPT-3 8-shot | 38.8 | 24.4 | 25.5 |
| GPT-3 16-shot | 40.5 | 18.9 | 23.3 |
| GPT-3 64-shot | 42.8 | 13.4 | 22.1 |
| **OOD Prompts w/ LM-Prob** | | | |
| TQA i.i.d. Prompt | 73.8 | 3.8 | 15.9 |
| NQ Prompt on TQA | 73.0 | 1.6 | 15.2 |
| DPR-BERT NQ $\rightarrow$ TQA | 33.1 | 33.1 | 35.2 |
| HQA i.i.d. Prompt | 29.8 | 25.0 | 23.5 |
| NQ Prompt on HQA | 27.7 | 24.1 | 25.2 |
| DPR-BERT NQ $\rightarrow$ HQA | 23.6 | 45.7 | 42.4 |

Table 6: Accuracy, ECE, and Brier scores of GPT-3 and the DPR-BERT baseline. GPT-3 is better calibrated than supervised DPR-BERT on both in-domain and OOD settings.

**Confidence Scoring** We compare two ways of estimating confidence for GPT-3 predictions. **LM-Prob**: the (normalized) language model probability, also equivalent to the reciprocal of perplexity, is $Conf \equiv P(w_1 w_2 \ldots w_n)^{\frac{1}{N}}$ where $w_1 w_2 \ldots w_n$ are the generated tokens in the answer. **Self-Con**: We also explore using self-consistency (Wang et al., 2023) to obtain confidence measures. Following Wang et al. (2023), during decoding we set a high temperature value (0.7) and sample 10 times for a set of different predictions. Among all the generated answers, we take the most frequent answer as the final prediction and its frequency as the confidence score.

---

[4] In Appendix Figure 2.

|       | DPR-BERT NQ | LM-Prob NQ | Self-Con NQ | LM-Prob TriviaQA | LM-Prob HotpotQA |
|-------|-------------|------------|-------------|------------------|------------------|
| 100%  | 36.1        | 40.5       | 40.2        | 73.8             | 29.8             |
| 50%   | 41.9        | 58.8       | 62.0        | 88.5             | 47.6             |
| 10%   | 60.1        | 83.1       | 77.0        | 95.4             | 68.1             |

Table 7: Selective prediction results. All numbers represent the accuracy (EM) at the corresponding coverage thresholds. For example, 100% means performance on the entire test set while 10% means the performance on the most confident 10% predictions. Both LM-Prob and Self-Con allow effective selective prediction with high accuracy on the most confident subsets. More results in Appendix D.

## 4.2 Results

While still imperfect, GPT-3 (with either LM-Prob or Self-Con) is better calibrated than supervised DPR-BERT (Table 6). Most calibration errors come from overconfidence where the predictions' confidence is higher than expected accuracy. Interestingly, while increasing the number of examples in the prompt improves accuracy, the calibration does not improve. For example, the 2-shot accuracy is 5.8 points worse than 64-shot but better calibrated. Moreover, while OOD transfer is a challenge for supervised models' calibration (tends to be overconfident on OOD test sets), GPT-3 has similar calibration regardless of the source of examples.

The selective prediction results show confidence scores can rank model predictions (Table 7): the most confident predictions have much higher accuracy. Moreover, GPT-3's confidence scores are more discriminative. For example, while the average accuracy on NQ is similar between GPT-3 and DPR-BERT, the top 10% predictions get an accuracy of 83.1% while for DPR-BERT it is only 60.1%. Such selective prediction can be very useful in practical settings, for example, we only trust the most confident predictions from the model and ask humans to verify the rest, making the use of GPT-3 more reliable.

**Takeaway** ($i$) Language model probability and self-consistency frequency can produce better calibration on GPT-3 than a supervised DPR-BERT model, especially on OOD test sets. ($ii$) Increasing the number of demos in the prompt improves accuracy but not necessarily calibration. ($iii$) We can perform effective selective prediction based on GPT-3 confidence scores.

## 5 Facet 4: Factuality Via Knowledge Updating

Although large language models store vast knowledge in their parameters (Petroni et al., 2019), the model is sometimes wrong or out of date, rendering them unreliable for knowledge-intensive tasks. In this section, we improve this factuality aspect of reliability by improving the prompting methods.

## 5.1 Memorization vs Updating

The larger a model, the more it can memorize (Carlini et al., 2023), this raises the concern of whether large models like GPT-3 can **forget** memorized knowledge when needed and **update** its knowledge.

**Experiment Setup** Our evaluation setup is inspired by Longpre et al. (2021), who reason about counterfactual scenarios. Specifically, we sample 36K and 18K questions from NQ and SQuAD's training splits (respectively, using the splits provided by MRQA). We use 16 demo examples from each dataset as the prompt for closed-book QA first. We assume that if GPT-3 gets the answer to the question right in the closed-book setting, then it has already memorized that piece of knowledge. We keep the set of questions where GPT-3 got right in the closed-book setting (for NQ, 21188 questions; for SQuAD, 7035 questions), and for these questions, we append a counterfactual passage supporting an alternative answer. We construct these counterfactual using the entity-swap from Longpre et al. (2021): for each question, take its gold passage and replace the gold answer entity with another entity with the same type sampled from the same QA corpus. After such entity substitution, the counterfactual passages support the substituted answer instead of the original answer. Our expectation is that the model should generate this updated answer given this counterfactual passage, instead of its original memorized answer. We randomly sample 16 demo examples as the prompt

|  | NQ | TriviaQA | SQuAD |
|---|---|---|---|
| DPR-BERT (supervised) | 41.5 | 56.8 | 24.1 |
| Atlas-11B (64-shot) | 42.4 | 74.5 | – |
| GPT-3 Closed-Book | 40.6 | 73.6 | 20.2 |
| + Contriever top-5 | 43.3 (61.8%) | 75.6 (69.6%) | 31.7 (48.8%) |
| + Contriever top-10 | 44.2 (70.5%) | 76.0 (75.1%) | 34.0 (57.7%) |

Table 9: GPT-3 16-shot prompting results on open-domain QA datasets. We use Exact Match as the metric. For Contriever results, we additionally show the retriever's recall in brackets. Adding retrieval to GPT-3 consistently improves QA accuracy.

and we use triples of the answer-substituted passage, the question, and the substitution answers ($\langle P'$, $Q, A' \rangle$) in the prompt to specify the task of performing reading comprehension based on the passage.

**Measuring How Well can GPT-3 Update its Knowledge** There are three possible outcomes: 1) the model retains the memorized answer; 2) the model predicts the updated answer (i.e., the substitution entity in the counterfactual passage); 3) the model predicts some other answer. We measure the proportion of those outcomes and hope models to update answers more often. For a baseline, we include results from Longpre et al. (2021): a fine-tuned T5 reader—trained on NQ and NewsQA—model with a DPR retriever.

**Results** As shown in Table 8, we find that when prompting with counterfactual triples ($\langle P', Q, A' \rangle$), GPT-3 can update about 85% of the time, much higher than the supervised baseline (Table 8). Comparing *Text-Davinci-001* and *Text-Curie-001*, the larger model also updates better to new answers in counterfactual passages.

|  | Retain$_\downarrow$ | Update$_\uparrow$ | Other$_\downarrow$ |
|---|---|---|---|
| *NQ* with *Code-Davinci-002* | | | |
| T5 (770M, supervised) | 20% | 33% | 47% |
| GPT-3 | 4.5% | 85.4% | 10.2% |
| *SQuAD* with *Code-Davinci-002* | | | |
| GPT-3 | 7.1% | 84.8% | 8.1% |
| *NQ* with different GPT-3 models | | | |
| *Text-Davinci-001* (175B) | 7.2% | 57.9% | 34.9% |
| *Text-Curie-001* (6.7B) | 14.8% | 40.0% | 45.2% |

Table 8: In-context knowledge updating results for memorized answers in NQ and SQuAD. When giving counterfactual examples in the prompt, GPT-3 updates its answers around 85% of the time, much higher compared to the supervised model. Moreover, larger models are better at in-context knowledge updating.

## 5.2 RETRIEVAL-AUGMENTED OPEN-DOMAIN QA

Large language models can answer closed-book QA from the model's stored knowledge (Roberts et al., 2020). However, a prompt can judiciously add more relevant information especially given our findings from the previous section that GPT-3 can update its knowledge with information in the prompt. We thus explore improving factual QA via retrieval-augmented prompts.

**Approach** We use the unsupervised Contriever model (Izacard et al., 2022a): for a test question, retrieve the top passages from the Wikipedia dump, concatenate them, and prepend them to the test question. Since the context is length-limited, we only prepend retrieved passages to the test question, not the demo examples, so the demo examples are only in the form of question-answer pairs. We compare this retriever-augmented approach with a closed-book baseline where we do not add the retrieved passages in the prompt. The demo examples used for both the retrieval-augmented prompting and closed-book prompting are exactly the same.

**Results** Adding retrieved passages into the prompt consistently boosts GPT-3 performance on all three open-domain QA datasets (Table 9), with particularly large gains on SQuAD (possibly because answers in SQuAD are spans from Wikipedia passages rather than free-form answers). Moreover, having better recall for retrieval gives better performance.

|  | Overall | Sub-Q1 | Sub-Q2 |
|---|---|---|---|
| Standard Prompting | 18.0 / 28.1 | 40.1 / 49.6 | 43.3 / 58.4 |
| CoT | 25.2 / 35.2 | 30.3 / 37.4 | – |
| CoT + Human Sub-Q1 | 30.0 / 42.3 | 44.2 / 54.1 | – |
| CoT + Human Sub-Q1 + Gold Sub-A1 | 44.3 / 59.0 | – | – |

Table 10: Results on HotpotQA as well as the decomposed sub-questions (we report EM / F1). Incorporating decomposed sub-questions in the prompt makes the model adjust its follow-up step predictions and significantly improves the overall answer accuracy.

## 5.3 REASONING-AUGMENTED MULTI-HOP QA

The above experiments demonstrate the effectiveness of ensuring GPT-3's factuality via in-context knowledge updating; however, it is mostly constrained on simple single-hop factual questions. In real-world applications, many user queries are multi-hop - they require multiple steps of reasoning over factual knowledge. Ensuring factuality in multi-hop questions involves additional challenges: models may fail because they derive the reasoning steps wrongly. To tackle this more challenging multi-hop setting, we study whether it is possible to improve GPT-3's multi-hop reasoning by incorporating human-written question decomposition in the prompt.

**HotpotQA and Decomposed Sub-Questions** We use the HotpotQA dataset (Yang et al., 2018) for our experiments, which consists of multi-hop questions that require at least two steps of reasoning. We use the question decomposition from Tang et al. (2021), where HotpotQA questions are annotated as decomposed (single-hop) sub-questions with corresponding intermediate answers.

**Baseline: Chain-of-Thought Prompting** Chain-of-Thought (CoT) prompting (Wei et al., 2022b) is a new prompting method tailored to multi-step questions, which we adopt in our experiments as a baseline, where we provide human-written reasoning steps for all demo examples to induce similar reasoning on test examples. We measure accuracy of GPT-3's final answer predictions on HotpotQA (Overall) as well as on the decomposed single-hop sub-questions. From the first row of Table 10, we see that standard prompting achieves higher accuracy on the single-hop sub-questions than the entire multi-hop questions as expected. CoT generates the entire reasoning chain along with its decomposed sub-questions and the intermediate answers to sub-questions, where the accuracy on the multi-hop questions is higher than standard prompting (second row of Table 10).

**Incorporating Human Decomposition** Instead of relying on GPT-3 itself to generate reasoning chains, we add the human-written question decomposition into the prompt. When adding the human-written sub-questions for the first step of reasoning (second last row of Table 10), we see a clear improvement in both the overall multi-hop QA accuracy as well as the sub-question accuracy. Moreover, when we further add the human-written QA pair of the first decomposed question in the reasoning chain (last row of Table 10), there is an even larger performance gain on the multi-hop QA performance. This shows that GPT-3 is able to adapt to the question decomposition information from humans and deduce the subsequent reasoning steps to eventually obtain the correct answers, offering better control and reliability.

**Takeaway** (*i*) Adding retrieved evidence passages can improve GPT-3 performance on factual QA. (*ii*) GPT-3 can update its knowledge when provided passages conflicting with its memorized knowledge. (*iii*) Incorporating human-written question decomposition corrects the reasoning chains of GPT-3 and improves performance on multi-hop QA.

## 6 CONCLUSION

Our work systematically studies the reliability of GPT-3 from four key facets: generalizability, fairness, calibration, and factuality. We develop effective prompting strategies to make GPT-3 outperform supervised models by large margins on these facets. Our work reveals new insights of LLMs and provides practical recommendations for users of GPT-3. We hope our work can inspire more future work to: (1) examine more facets of reliability, such as avoiding harmful generations; (2) apply the prompting methods in this paper to more real-world applications, such as incorporating human feedback for collaborative multi-step planning; (3) further explore more effective prompting strategies to improve reliability, such as post-hoc calibration on language model probabilities.

## ETHICAL STATEMENT

**Ethical Use of GPT-3**    The goal of this project is to avoid the potential harm of GPT-3 and all of our GPT-3 experiments are motivated to better study and improve reliability. We believe our experiments and findings can improve the reliability and allow safer use of the model. In particular, our section on social biases and fairness is a key aspect of the ethical use of GPT-3. We presented evidence that the model exhibits biased predictions, especially when the demo examples in the prompt have a skewed demographic distribution. Although we explored ways of mitigating these biases, the model is still far from perfect, and there is much more work needed to further improve its fairness. We take our work as an initial step towards more ethical use of GPT-3.

**Limitations of This Work**    We note several limitations of this work and suggest a list of open questions for future work.

- **Other reliability facets**: In this work, we covered four key facets of reliability, but there are surely other facets that we may have missed. For example, combatting adversarial examples identified via human or AI red-teaming (Ganguli et al., 2022; Branch et al., 2022; Perez et al., 2022), detecting and handling malicious prompts such as prompt injection [5], and avoiding toxic and hallucinated generations (Gehman et al., 2020; Gao et al., 2022).

- **Methods for improving reliability**: Although we have taken initial steps and discovered some effective prompting strategies for these reliability facets, readers should not take this work as evidence that GPT-3 is already reliable and ready for deployment. In fact, our experiments indicate ample room for further improvement, for example in reducing social biases and improving calibration. We hope this work inspires more future work that develops more effective strategies to make LLMs reliable.

- **Analysis to understand model behaviors**: While we have found interesting properties of GPT-3, it remains unclear what exactly caused these behaviors. For example, if the small generalization gap due to the use of prompting, or the training data, or the training objectives or model architecture? When GPT-3 is sensitive to the prompt in debiasing, is it triggered by certain keywords or phrases? Why ordering anti-bias examples at the end of the prompt does not lead to the recency bias (Zhao et al., 2021b) but rather still incurs strong biases against minority groups? Can we attribute model behaviors to the pretraining data or interpret model attention patterns? These analysis can potentially help us better understand how and why prompting works and therefore allow us to better leverage LLMs.

## ACKNOWLEDGMENT

We thank Jason Phang, Ziyi Yang, Dan Friedman, Sewon Min, Jieyu Zhao, He He, Alicia Parrish, Chen Zhao, Shi Feng, Han Guo, Weijia Shi, Jungo Kasai, Xi Ye, Su Lin Blodgett, Trista Cao, Ekin Akyürek, Leo Boytsov, Aishwarya Kamath, Weijia Xu, Yankai Lin, Xiaozhi Wang, Zhengyan Zhang, and many other friends from UMD CLIP and the Azure AI team at Microsoft for their helpful discussion and feedback.

---

[5]https://simonwillison.net/2022/Sep/12/prompt-injection/

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

APPENDIX

## A  MORE RELATED WORK

**Robustness to Distribution Shifts.**  Machine learning models are known to overfit their training distribution and often suffer performance degradation when the test distribution differs from the training distribution. In the case of language models, various forms of distribution shifts have been studied. For example, domain shifts pose great challenges for LLMs on question answering (Talmor & Berant, 2019; Fisch et al., 2019) and text classification (Hendrycks et al., 2020; Arora et al., 2021); various forms of adversarial attacks can break LLMs even by just strategic synonym substitution, paraphrase, or distractor insertion (Jin et al., 2020; Li et al., 2020; Ribeiro et al., 2018; Si et al., 2019; 2021a; Jia & Liang, 2017; Si et al., 2021b); LLMs have been shown to exploit shortcuts or spurious correlations in the training data which fail on counter test examples (McCoy et al., 2019; Zhang et al., 2019; Poliak et al., 2018; Gururangan et al., 2018); LLMs also fail on new compositional structures that are not observed during traing (Kim & Linzen, 2020; Keysers et al., 2020; Bogin et al., 2022). In real-world settings, various forms of distribution shifts can happen and reliable models should perform well even when encountering such out-of-distribution (OOD) examples. Intuitively, in-context few-shot prompting should suffer less OOD degradation since the pretrained parameters are preserved, unlike the case of supervised finetuning. We perform a series of empirical evaluations on domain shift, curated challenge sets, and spurious correlation to validate this hypothesis.

**Bias and Fairness.**  Language models producing toxic or biased content can cause severe harm especially to the groups being biased against (Bender et al., 2021). A series of benchmarks have been developed to show that LLMs can generate toxic outputs (Gehman et al., 2020), contain gender biases (Rudinger et al., 2018; Zhao et al., 2018) and other categories of social biases (Nangia et al., 2020; Nadeem et al., 2021; Parrish et al., 2022), perform poorly against minority demographic groups (Koh et al., 2021; Harris et al., 2022) or dialectical variations (Ziems et al., 2022; Tan et al., 2020). Ideally, LLMs should not exhibit biased behaviors and not discriminate against any group. While many of these evaluations focus on evaluating the internal representation of LLMs in a zero-shot setting or evaluating the biases on specific downstream applications in a supervised setting, it remains unclear how these biases change under different prompting schemes in the few-shot setting, which will be the focus of our analysis. A closely related work is Lucy & Bamman (2021) which study representation biases in GPT-3 generated stories. We instead evaluate on the downstream tasks of coreferece resolution and question answering. Apart from few-shot prompting, Solaiman & Dennison (2021) proposed a general method to align language models with human values, but it involves expensive iterative training.

**Uncertainty Calibration.**  No model can ever be perfect, and so it is crucial for users to be able to identify model mistakes, especially in high-stage settings where trusting wrong model predictions can cause severe harm. One important way to help identify wrong model predictions is by obtaining well-calibrated confidence scores for model predictions. By definition, a calibrated confidence (probability) score should match the expected accuracy of the prediction (Platt, 1999; Naeini et al., 2015; Guo et al., 2017). In this way, users can put more trust in highly-confidence predictions and discard low-confidence predictions. While various methods have been proposed to obtain confidence scores and perform post-hoc calibration for language models (Jiang et al., 2021; Desai & Durrett, 2020; Ye & Durrett, 2022), they are mostly focused on classification settings rather than free-form generation, which is more common for the use of GPT-3. In this work, we explore two simple (but surprisingly effective) ways of obtaining confidence scores for GPT-3's generated answers and we analyse the impact of scaling as well as prompt design. For studying calibration of GPT-3 style LLMs, Lin et al. (2022a) explore the idea of expressing uncertainty in verbal words but is restricted to math questions. Mielke et al. (2022) study linguistic calibration on conversational models. Kadavath et al. (2022) study adopting a multiple-choice setting in which case obtaining a confidence score is much easier (since the model only needs to predict one token to indicate which option to choose rather than generating the entire answer string). We differ from them in: 1) we focus on obtaining probabilistic confidence scores rather than verbal uncertainty expressions; 2) we study the more general and realistic free-form answer generation setting; and 3) we do not involve finetuning or any additional training of the language model.

|  | SQuAD | HotpotQA | TriviaQA | NewsQA | SearchQA | NQ | Average |
|---|---|---|---|---|---|---|---|
| D-Net | – | – | – | – | – | – | 84.1 |
| Delphi | – | – | – | – | – | – | 82.3 |
| MultiFT | 91.8 | 81.0 | 80.1 | 72.3 | 84.7 | 79.5 | 81.6 |
| MADE | 91.9 | 80.7 | 80.1 | 71.8 | 84.5 | 79.5 | 81.4 |
| T5-Finetune | 94.9 | – | – | – | – | – | – |
| T5-PromptTune | 94.8 | – | – | – | – | – | – |
| GPT-3 Source-P | 87.8 | 78.9 | 88.6 | 60.1 | 87.3 | 76.2 | 79.8 |

Table 11: Results on MRQA in-domain datasets. We use the F1 metric for all datasets. D-Net and Delphi results are taken from the MRQA system report (Fisch et al., 2019) which only reported the average performance. MultiFT and MADE results are taken from Friedman et al. (2021). T5-Finetune and T5-PromptTune results are from the prompt tuning paper (Lester et al., 2021) which only trained on the SQuAD datsaet. GPT-3 few-shot performance slightly lags behind these supervised baselines in the in-domain setting.

**Knowledge Updating.**    Despite the fact that LLMs like GPT-3 are pretrained on very large corpora, they are still far from perfect in terms of factual knowledge. On one hand, they still make factual mistakes even on domains that have seen before during pretraining (e.g., Wikipedia); on the other hand, they are pretrained on static corpora and hence their knowledge can become outdated. In order for LLMs to serve as reliable knowledge bases (Petroni et al., 2019) or power knowledge-intensive downstream applications (Petroni et al., 2021), it is important to keep LLMs' knowledge factually correct and up-to-update. A recent line of work attempts to edit factual knowledge in LLMs by making targeted modifications of the model's neurons (Cao et al., 2021; Mitchell et al., 2021; 2022). However, these methods are hard to be applied on GPT-3 since it is much larger in size and often treated as a black box without access to internal parameters. To address this issue, in this paper we explore the feasibility of performing in-context knowledge updating by directly appending relevant knowledge pieces in the prompt to guide model predictions. Since it has been shown that larger models are better at memorization (Carlini et al., 2023), we analyze whether it is possible to make larger models forget their memorized knowledge and adapt to the new information presented in the prompt, especially when these two are in conflict. The idea of adding retrieved passages is conceptually similar to the line of work on retrieval-augmented methods for knowledge-intensive NLP (Lewis et al., 2020; Izacard et al., 2022b; Guu et al., 2020). However, these methods still require supervised training while we focus on the setting of few-shot prompting with all the language model's parameters being frozen.

## B    ADDITIONAL RESULTS: GENERALIZABILITY

We provide full experimental results and comparisons with more baselines.

**MRQA**    Table 11 and Table 12 present detailed results on MRQA. For baselines, we include results from the top-performing systems of the MRQA competition: D-Net (Li et al., 2019) and Delphi (Longpre et al., 2019), a recent adapter-based robust tuning method MADE (Friedman et al., 2021) as well as their multi-dataset finetuning baseline. We also report the finetuning and prompt tuning Lester et al. (2021) result of using T5, which achieves state-of-the-art OOD transfer results on MRQA. Note that this T5 baseline only uses SQuAD as the in-domain training data.

**AdvGLUE and Contrast Sets**    Table 13 and Table 14 present detailed results on AdvGLUE and Contrast Sets, where GPT-3 shows better generalization than supervised baselines.

**Spurious Correlation**    Table 16 shows full the performance breakdown on HANS based on the three spurious features. We can see that for the subsequence and constituent features in HANS, GPT-3 still suffers significant performance gaps between the bias-supporting and bias-countering subsets. This leaves curious questions like why such gaps only occur for certain bias features but not others, and how such spurious biases arise (most likely due to pretraining), and we leave a more thorough analysis of these questions to future work.

|  | BioASQ | DROP | DuoRC | RACE | RE | TextbookQA | Average |
|---|---|---|---|---|---|---|---|
| D-Net | – | – | – | – | – | – | 69.7 |
| Delphi | – | – | – | – | – | – | 68.5 |
| MultiFT | 64.1 | 51.5 | 63.0 | 47.6 | 87.3 | 59.0 | 62.1 |
| MADE | 66.5 | 50.9 | 67.2 | 47.8 | 86.7 | 58.5 | 62.9 |
| T5-Finetune | 77.9 | **68.9** | 68.9 | 59.8 | 88.4 | 54.3 | 69.7 |
| T5-PromptTune | 79.1 | 67.1 | 67.7 | 60.7 | 88.8 | 66.8 | 71.7 |
| GPT-3 Source-P | **86.2** | 67.7 | **70.5** | **69.0** | 89.3 | **84.8** | **77.2** |
| GPT-3 Target-P | 85.9 | **68.9** | 69.7 | 65.4 | **91.0** | 82.1 | **77.2** |

Table 12: Results on MRQA OOD datasets. ID-P (second last row) uses a fixed set of sampled demo examples from the in-domain datasets and evaluates on these OOD datasets, while OOD-P (last row) uses sampled demo examples from each of these OOD datasets for evaluation. GPT-3 significantly outperforms all other supervised baselines on these OOD datasets, moreover, using the in-domain prompt successfully transfers to OOD test data, achieving the same average performance as using demos examples drawn from these OOD datasets as the prompt.

|  | SST-2 | MNLI | RTE | QNLI | QQP | Average |
|---|---|---|---|---|---|---|
| | *Clean Test Sets* | | | | | |
| RoBERTa | 96.0 | 89.8 | 86.6 | 94.1 | 92.0 | 91.7 |
| ALBERT | 95.2 | 89.6 | 88.4 | 95.3 | 92.3 | 92.2 |
| DeBERTa | 96.3 | 90.9 | 90.2 | 94.9 | 92.3 | 92.9 |
| GPT-3 | 96.1 | 78.1 | 83.4 | 79.8 | 83.5 | 84.2 |
| | *Adversarial Test Sets* | | | | | |
| RoBERTa | 58.5 | 45.2 | 45.4 | 52.5 | 57.1 | 51.7 |
| ALBERT | 66.8 | 48.0 | 73.0 | **63.8** | 56.4 | 61.6 |
| DeBERTa | 57.9 | 55.5 | 78.9 | 57.9 | 60.4 | 62.1 |
| GPT-3 | **78.4** | **56.4** | **87.7** | 57.4 | **66.7** | **69.3** |

Table 13: Results on the clean and adversarial test sets of AdvGLUE, we use accuracy as the metric for all datasets. For MNLI, we report the average performance on the matched and mismatched dev sets. The supervised baselines are trained on the clean training data, and GPT-3 uses few-shot prompts sampled from the clean training data. While GPT-3 few-shot prompting lags behind supervised models on clean test sets, it significantly outperforms the supervised models on the adversarial sets. That being said, we still note a performance drop of GPT-3 on the adversarial test sets when using clean demo examples.

In Table 15, we perform additional ablation on the impact of the number of demos and the different GPT-3 variants. With fewer demo examples from QQP, despite a slight drop on the QQP test set, GPT-3 actually remains robust (even higher accuracy on PAWS than the 16-shot results). On the other hand, using the Text-Davinci-001 (175B) and the smaller Text-Curie-001 (6.7B) performs far worse on both the QQP test set and the PAWS challenge test set.

## C   ADDITIONAL RESULTS: SOCIAL BIASES

We also break down the accuracy and bias scores of using different prompts in Table 17 by the different bias categories. We observe that there can be large differences across different categories. Moreover, we underlined the categories from which the demo examples come, and we observe that having same-category demos in the prompt does not correlate with the bias scores. For instance, we have bias-supporting examples from the Nationality category in the Ambig-Pro case but the bias score remains low, while the bias score for the Physical Appearance and Disability categories becomes much higher even when the biased examples are not from these categories.

|  | IMDB - Original | IMDB - Contrast | Gap↓ |
|---|---|---|---|
| BERT | 93.8 | 84.2 | 9.6 |
| GPT-3 | 94.1 | 93.6 | 0.5 |
|  | QuoREF - Original | QuoREF - Contrast | Gap↓ |
| XLNet | 70.5 | 55.4 | 15.1 |
| GPT-3 | 86.1 | 77.0 | 9.1 |
|  | BoolQ - Original | BoolQ - Contrast | Gap↓ |
| RoBERTa | 86.1 | 71.1 | 15.0 |
| GPT-3 | 85.5 | 80.0 | 5.5 |

Table 14: Results on Contrast Sets. For IMDB and BoolQ, we report accuracy; for QuoREF, we report F1. Apart from the performance on the original and contrast sets of the three datasets, we also note the gap between performance on the original and contrast sets. We see a clear trend that GPT-3 incurs a smaller gap than the supervised models.

|  | QQP (ID) | PAWS (OOD) | Gap |
|---|---|---|---|
| BERT (supervised) | 91.3 | 40.1 | 51.2 |
| RoBERTa (supervised) | 89.0 | 39.5 | 49.5 |
| Code-Davinci-002 (4-shots) | 78.2 | 80.5 | -2.3 |
| Code-Davinci-002 (16-shots) | 83.5 | 73.7 | 9.8 |
| Text-Davinci-001 (16-shots) | 72.4 | 42.4 | 30.0 |
| Text-Curie-001 (16-shots) | 40.1 | 32.1 | 8.0 |

Table 15: Ablation for the impact of the number of demos and different GPT-3 model variants on MNLI-HANS.

## D  ADDITIONAL RESULTS: CALIBRATION

The full selective prediction results in Table 18 show that the confidence scores can be used to rank model predictions. We see a clear trend that the most confident predictions have much higher accuracy.

The reliability diagrams in Figure 2 show that in most cases the calibration errors come from overconfidence where the predictions' confidence is higher than the expected accuracy. It is also worth noting while OOD transfer is a big challenge for the calibration of supervised models where there tends to be overconfidence on the OOD test sets, GPT-3 exhibits similar calibration results when using in-domain or OOD demo examples as the prompt (bottom-left plot in Figure 2).

To further disentangle the impact of better accuracy and better calibration, we perform a controlled evaluation of selective prediction in Table 19 where we sub-sample the NQ test set so that the three calibration methods achieve the same accuracy on the test set. We see a clear trend that despite DPR-BERT and GPT-3 get same accuracy on this sub-sampled test set, DPR-BERT gets much higher accuracy on the most confident predictions indicating the usefulness of better calibration.

## E  ADDITIONAL RESULTS: KNOWLEDGE UPDATING

### E.1  IMPACT OF PROMPTS FOR MEMORIZATION VS UPDATING

For knowledge updating, we compare several different prompt designs as detailed below, for all cases, we randomly sample 16 demo examples as the prompt.

- $\langle Q, A \rangle$ : We use the original question-answer pairs in the prompt.
- $\langle P, Q, A \rangle$ : We use the original passage-question-answer triples in the prompt (i.e., the answer in the passage remains the original gold answer).

| HANS Category | GPT-3 Acc. |
|---|---|
| Lexical Overlap - Entailment | 87.9 |
| Lexical Overlap - Non-Entailment | 96.9 |
| Subsequence - Entailment | 84.0 |
| Subsequence - Non-Entailment | 53.7 |
| Constituent - Entailment | 87.2 |
| Constituent - Non-Entailment | 42.2 |

Table 16: Breakdown of GPT-3 results on HANS by categories. All demo examples in the prompt are from MNLI. GPT-3 still suffers significant performance gaps between the entailment (bias-supporting) and non-entailment (bias-countering) subsets for the subsequence and constituent features in HANS.

| | Balanced | | Ambig-Pro | | Ambig-Anti | |
|---|---|---|---|---|---|---|
| | Acc$_\uparrow$ | Bias$_\downarrow$ | Acc$_\uparrow$ | Bias$_\downarrow$ | Acc$_\uparrow$ | Bias$_\downarrow$ |
| SES | 96.4 / 74.2 | 3.2 / 0.0 | 14.2 / 99.2 | -12.6 / 0.0 | 8.4 / 99.6 | -6.4 / 0.0 |
| Sexual orientation | 97.4 / 76.3 | 1.6 / 1.2 | 1.9 / 98.4 | 19.1 / -0.9 | 2.8 / 97.2 | 17.2 / -0.9 |
| Religion | 97.0 / 75.8 | 1.8 / 0.5 | 0.4 / 98.0 | 35.2 / 1.2 | 1.0 / 97.2 | 24.6 / 1.6 |
| Race | 99.8 / 85.1 | -0.1 / -0.5 | 1.2 / 99.0 | 4.0 / 0.1 | 2.7 / 98.9 | 3.7 / 0.1 |
| Physical Appearance | 97.4 / 56.0 | 2.6 / 18.8 | 1.4 / 87.8 | 75.0 / 14.8 | 0.6 / 86.2 | 77.0 / 14.8 |
| Nationality | 98.2 / 80.8 | 1.4 / -11.6 | 1.0 / 99.0 | -0.2 / 0.0 | 1.0 / 98.6 | 0.6 / 0.0 |
| Gender identity | 99.0 / 66.8 | 0.6 / -3.9 | 6.0 / 98.6 | 5.6 / 0.4 | 4.6 / 98.8 | 3.8 / 0.4 |
| Disability | 97.4 / 74.2 | 2.2 / 8.5 | 0.0 / 96.6 | 85.2 / 6.0 | 0.2 / 97.2 | 82.6 / 4.8 |
| Age | 82.2 / 76.6 | 13.0 / 8.1 | 0.0 / 95.4 | 52.0 / 12.4 | 0.4 / 95.6 | 48.4 / 12.0 |

Table 17: Breakdown of accuracy and bias score results on BBQ. For accuracy and bias scores, the first number represents the ambiguous set and the second number represents the disambiguated set. Underlined numbers indicate that there are demo examples in the prompt from the same bias category.

- $\langle Q, A' \rangle$ : We use the question-answer pairs, but with the substitution entities as gold answers in the prompt.
- $\langle P', Q, A' \rangle$ : We use triples of the answer-substituted passage, the question, and the substitution answers in the prompt.

As shown in Table 20, we find that the prompt design has a big impact on the knowledge updating behavior. In particular, showing only the original passage-question-answer triples ($\langle P, Q, A \rangle$) still causes high memorization ratios, however, when prompting with counterfactual triples ($\langle P', Q, A' \rangle$), GPT-3 can update 85% of the time with much lower memorization ratios than a supervised model.

## E.2 TARGETED IN-CONTEXT KNOWLEDGE UPDATING

The experiments in the previous section showed promise that GPT-3 can adapt to new knowledge given in the prompt when there is a conflict with its memorized knowledge. One missing aspect from the above analysis is whether we can perform **targeted** knowledge update: when given a piece of knowledge update, we expect the model to predict the updated answer for all questions related to that knowledge, but not change its answer for other unrelated questions. To assess model behavior on this front, we adopt an evaluation setup closer to recent knowledge updating literature (Cao et al., 2021; Mitchell et al., 2021).

**Experiment Setup** We use two evaluation datasets from Mitchell et al. (2021): 1) We first use the fact checking dataset FEVER (Thorne et al., 2018): each claim requires a binary true / false judgement. We create the edited label which is opposite to the originally predicted label from GPT-3. For example, for a test example, if the original GPT-3 prediction is true, then the new label for editing would be false. We present the knowledge update in the form of a natural sentence that supports the target label for editing. We then test whether GPT-3 predicts the target label for a

|       | DPR-BERT NQ | LM-Prob NQ | Self-Con NQ | LM-Prob TriviaQA | LM-Prob HotpotQA |
|-------|-------------|------------|-------------|------------------|------------------|
| 100%  | 36.1        | 40.5       | 40.2        | 73.8             | 29.8             |
| 90%   | 38.0        | 43.7       | 44.3        | 78.3             | 32.7             |
| 80%   | 39.5        | 46.8       | 48.7        | 81.7             | 36.0             |
| 70%   | 40.6        | 50.2       | 53.1        | 84.1             | 39.7             |
| 60%   | 41.2        | 53.7       | 57.8        | 86.5             | 43.5             |
| 50%   | 41.9        | 58.8       | 62.0        | 88.5             | 47.6             |
| 40%   | 43.3        | 63.3       | 66.0        | 90.5             | 52.1             |
| 30%   | 46.1        | 70.2       | 71.2        | 92.5             | 56.5             |
| 20%   | 49.2        | 77.4       | 74.7        | 93.7             | 61.6             |
| 10%   | 60.1        | 83.1       | 77.0        | 95.4             | 68.1             |

Table 18: Selective prediction results. All numbers represent the accuracy (EM) at the corresponding coverage thresholds. For example, 100% means performance on the entire test set while 10% means the performance on the most confident 10% predictions.

|       | DPR-BERT NQ | LM-Prob NQ | Self-Con NQ |
|-------|-------------|------------|-------------|
| 100%  | 35.0        | 35.0       | 35.0        |
| 90%   | 37.6        | 37.9       | 38.2        |
| 80%   | 39.0        | 40.8       | 42.8        |
| 70%   | 40.9        | 44.1       | 46.1        |
| 60%   | 41.6        | 47.8       | 51.6        |
| 50%   | 43.1        | 52.9       | 55.1        |
| 40%   | 44.1        | 58.0       | 60.6        |
| 30%   | 45.4        | 65.3       | 63.5        |
| 20%   | 49.5        | 75.3       | 71.5        |
| 10%   | 59.2        | 82.5       | 70.5        |

Table 19: Selective prediction results on a controlled set sampled from the original test set so that the accuracy on the sub-sampled test set is the same.

**paraphrase** of the original test claim. We measure accuracy on these paraphrases as the editing **success rate**. We sample a same-sized set from the training data that do not overlap with the test set as the set of unrelated questions. The intended behavior is that adding knowledge updates in the prompt does not impact performance on these unrelated questions. We measure performance drop on this unrelated set after and before adding knowledge updates as the accuracy **drawdown**. 2) We also use the zsRE question-answering dataset (Levy et al., 2017). For each test question, the target label for editing is randomly sampled from predictions by a smaller QA model which is different from the original GPT-3 prediction. Similarly, we measure accuracy on paraphrased test questions as success rate, and accuracy drop on a set of randomly sampled non-overlapping training questions as the accuracy drawdown.

**Prompt Design**   We compare several prompt designs (in particular what types of demo examples to use). For all cases, we sample 16 demos to use in the prompt.

- Original Examples Only: We only sample the original QA pairs (without editing information).
- Original + Edited Relevant Examples: We include demos examples for both original QA pairs as well as for questions with edited answers.
- Original + Edited Relevant + Edited Irrelevant Examples: We include demo examples covering all possible cases: the original QA pairs, QA pairs with knowledge update and updated answer, as well as QA pairs with knowledge update but original answer (where the question is unrelated to the knowledge update).

**Results**   From Table 21, we see that different prompts give vastly different results. Specifically, using only the original examples in the prompt leads to relatively poor success rate (especially on

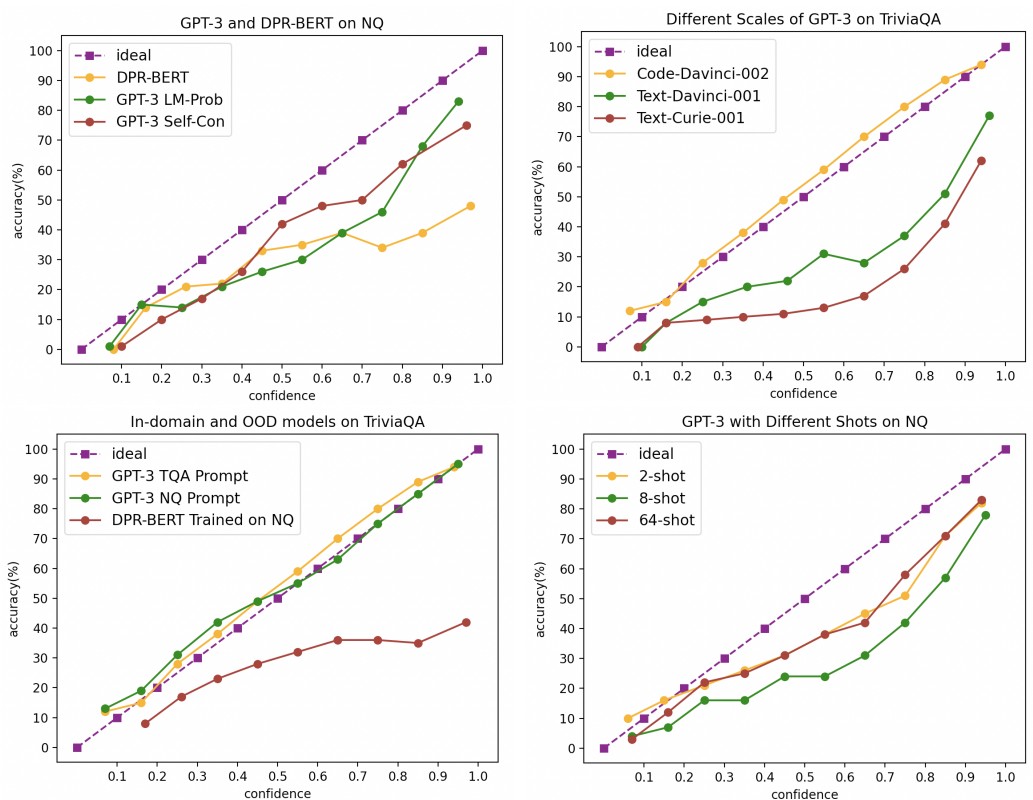

Figure 2: Reliability diagrams for different calibration setups. x-axis: confidence of each bucket; y-axis: accuracy of each bucket.

|  | Retain | Update | Other | Memorization Ratio |
|---|---|---|---|---|
| *NQ* with *Code-Davinci-002* | | | | |
| T5 (supervised) | 20% | 33% | 47% | 30% |
| GPT-3 Prompt $\langle Q, A \rangle$ | 60.8% | 25.8% | 13.4% | 70.2% |
| GPT-3 Prompt $\langle P, Q, A \rangle$ | 59.1% | 25.4% | 15.5% | 70.0% |
| GPT-3 Prompt $\langle Q, A' \rangle$ | 10.4% | 56.6% | 32.9% | 15.6% |
| GPT-3 Prompt $\langle P', Q, A' \rangle$ | 4.5% | 85.4% | 10.2% | 5.0% |
| *SQuAD* with *Code-Davinci-002* | | | | |
| GPT-3 Prompt $\langle Q, A \rangle$ | 58.0% | 29.1% | 12.9% | 66.6% |
| GPT-3 Prompt $\langle P, Q, A \rangle$ | 32.8% | 52.9% | 14.3% | 38.2% |
| GPT-3 Prompt $\langle Q, A' \rangle$ | 15.4% | 48.1% | 36.5% | 24.3% |
| GPT-3 Prompt $\langle P', Q, A' \rangle$ | 7.1% | 84.8% | 8.1% | 7.8% |
| *NQ* with different versions of GPT-3 | | | | |
| *Text-Davinci-001* (175B) | 7.2% | 57.9% | 34.9% | 11.0% |
| *Text-Curie-001* (6.7B) | 14.8% | 40.0% | 45.2% | 26.9% |

Table 20: In-context knowledge updating results for memorized answers in NQ and SQuAD. When giving counter-factual demo examples in the prompt, GPT-3 can update its answers around 85% of the time with low memorization ratio (as compared to supervised models). Moreover, we find that larger models are better at in-context knowledge updating.

FEVER), while adding edited relevant examples in the prompt leads to better success rate, it leads the model to over-rely on the knowledge updates even on irrelevant questions. However, when

|  | Success Rate (Relevant Paraphrases) | Acc. Drawdown (Irrelevant Questions) |
|---|---|---|
| Prompt: Original Examples Only | | |
| FEVER | 44.2 | 85.1 - 84.9 = 0.2 |
| zsRE QA | 92.9 | 40.0 - 39.7 = 0.3 |
| Prompt: Original Examples + Edited Relevant Examples | | |
| FEVER | 100.0 | 83.9 - 48.6 = 35.3 |
| zsRE QA | 99.9 | 39.7 - 11.6 = 28.1 |
| Prompt: Original Examples + Edited Relevant Examples + Edited Irrelevant Examples | | |
| FEVER | 99.9 | 84.0 - 83.5 = 0.5 |
| zsRE QA | 98.8 | 40.6 - 40.1 = 0.5 |

Table 21: Targeted in-context knowledge updating results for FEVER and zsRE. From the last block of results, we see that when using a mixture of all three types of demo examples, GPT-3 is able to achieve very high knowledge edit success rate (99.9% and 98.8%) while incurring minimal drawdown (0.5%) on irrelevant questions.

incorporating all cases of original examples, edited relevant and irrelevant examples in the prompt, GPT-3 is able to achieve high editing success rate and low drawdown on irrelevant questions.

