# OpenReview forum: "Prompting GPT-3 To Be Reliable"
_ICLR.cc/2023/Conference — ICLR 2023 poster_

### Official Review · Reviewer_d2Hh · 2022-10-25

**Confidence:** 3
**Correctness:** 3
**Technical Novelty And Significance:** 2
**Empirical Novelty And Significance:** 2
**Recommendation:** 6

**Clarity, Quality, Novelty And Reproducibility:**



**Clarity & quality:** the paper is well written. Few small typos (below) do not affect the overall quality. If I had to cram 4 research papers in 9 pages, I would probably do much worse.

**Novelty:** the paper has a large number of assorted finding in the four facets of reliability. To the best of my knowledge, at least some of them are not covered by related work, at least in their specific formulation with LLM prompting. Unfortunately, I am not an expert in all four research directions present in the paper and may have missed something (hence my confidence score).


### Minor stuff

> P1:  and their evaluation methods may not feasible on the much larger GPT-3 model.

May not BE feasible. Also, if you decide to keep the current direction of the paper, please explain why they are not feasible (possibly later in the paper).

> S3:   Random shuffling achieves the lowest bias gaps (...) Interestingly, putting either Pro-Bias or Anti-Bias examples at the end leads to much larger bias gaps. This contradicts the recency bias …

[mostly nitpicking] This contradiction was not clear to me. If you randomly order sentences, then, assuming recency bias, the most recent sentence is equally likely to be biased either way, and I thought it would average to zero. Perhaps it would be best to explain this contradiction in more detail.


**Strength And Weaknesses:**

### Strengths

1. The paper studies multiple highly impactful issues; As LLM prompting is gaining new applications, it is important to understand whether or not we can "trust" it - and how to make it more trustworthy.
2. Authors report a metric ton of evaluations. If the paper's worth was measured by the "total weight" of all contributions, this one would be among the best.
3. Authors make considerable effort to make the paper easier to follow. The main sections ("facets") follow a consistent structure. Each has a short summary at the end.
4. The paper is well-written and reasonably "tidy" -- especially if we take into account the density of information.
5. Authors clearly state the limitations of their work, both in the main paper and in the dedicated limitations section.

### Weaknesses

In general, it feels like the paper attempts to squeeze too much information into a limited space. Each of the four "facets" is an established research area with its own complexity, caveats, and prior work. It feels like the paper tries to be both an overview of these facets *and* an independent analysis of each one. In each specific section (facet), the paper has area-specific flaws: inconsistent evaluation, incomplete ablation, missing some task-specific caveats


1. missing comparisons and related work in individual facets, e.g. GPT-3 social / ethical issues [1,2,3], GPT-3 uncertainty[4], GPT-3 domain adaptation[5], knowledge editing[6], retrieval-augmented question answering [7, 8].
   * for instance, in Section 5 (knowledge updating), it is unclear how the retrieval-based editing compares with [6]  -- and which of the results are novel compared to [7, 8], and how does the proposed calibration strategy compare with [4].
   * please do not consider [1-8] as an exhaustive list of related work. I am not an expert in most of these facets. My point is: since a non-expert (me) can name multiple missing baselines, authors are likely to find more relevant work if they deliberately search for it.

2. As an overview of reliability: the choice of four specific facets feels arbitrary: e.g. it is unclear why we focus on generalizability and not, say, robustness to adversarial [9,10,11], or prioritize mitigating social bias over avoiding toxicity[2,12].

4. (Most) results are presented as-is, with no analysis. For instance, in Section 2, GPT-3 has a smaller generalization gap than a fine-tuned MLM model -- but what exactly causes this behavior?
  * Example A: Is this because of prompting v.s. fine-tuning? -> What if we use prompting for the MLM model?
  * Example B: Is this because of language model's training objective?  -> what if we compare similarly-sized CLM and MLM, e.g. T5-11B vs BLOOM/BLOOM-13B -- and try prompt tuning in the baseline model
  * Example C: Is this because of the training data? -> What if we compare CLMs and MLMs trained on similar *public* datasets?
  * Note: it is not a weakness that authors did not perform every possible analysis - but it would help the paper


5. (Some) claims are overly broad, e.g. "Language model probability and self-consistency frequency can produce better calibration than a supervised DPR-BERT model, specially on OOD test sets" -- this claim requires testing multiple language models in different setups.

6. It is not always clear how authors selected the baseline models for each task, e.g. S2 only has roberta, S3 adds deberta, S4 only bert, S5 is only T5

7. In T7, authors evaluate uncertainty calibration through selective prediction (e.g. accuracy on top-10% most confident). However, it is difficult to assess: what portion of the selective accuracy gain is due to better uncertainty calibration, and what portion is simply a more accurate base model. Prior art typically uses additional uncertainty metrics to decouple these two effects.



[1] https://aclanthology.org/2021.nuse-1.5/

[2] https://cdn.openai.com/palms.pdf

[3] https://aclanthology.org/2021.acl-long.416/

[4] https://arxiv.org/abs/2205.14334

[5] https://arxiv.org/abs/2112.08786

[6] https://arxiv.org/abs/2110.11309

[7] https://arxiv.org/abs/2005.11401

[8] https://arxiv.org/abs/2206.06520

[9] https://simonwillison.net/2022/Sep/12/prompt-injection/

[10] https://arxiv.org/abs/1908.07125

[11] https://arxiv.org/abs/2209.02128

[12] https://arxiv.org/abs/2112.04359







**Summary Of The Paper:**

This paper analyzes the "reliability" of large language models (LLM) with prompt engineering or in-context learning.  Authors consider four "facets" of reliability: (1) robustness to domain shift, (2) neutrality w.r.t. social biases (3) uncertainty calibration and (4) ability to update the learned knowledge. For each facet, authors study how LLM prompt-tuning / in-context learning compares against traditional fine-tuning of models such as RoBERTa, T5, DeBERTa, etc. For some facets, authors compare several prompting strategies in terms of their reliability. Aside from this general line of thought, the paper also contains a plethora of other results in each of the four respective "facets".

**Summary Of The Review:**



This paper combines 4 related (but separate) short research papers and an overview. To the best of my knowledge, all four research directions could be upgraded to a standalone short/full paper, through the addition of baselines, missing related work, fixing some task-specific issues and providing further justification of the already claimed results. At least 2 of 4 papers (and likely all 4) would stand a good chance to be accepted here or on *ACL / EMNLP-type conferences. The overview part has its own merit: should authors modify it to add relevant prior work and focus on analyzing existing results, it would be a valuable resource to beginners.

However, in the current form, each of the four research directions is only given a couple of pages, lacking background and task-specific nuances - and adding this information would be difficult because of the size limit, and deferring it to the appendix would make the paper difficult to follow. I would consider one of two strategies:

* splitting the paper: giving each research case at least a short standalone paper -- where you will have space to better support your main claims.
* reformatting as a meta-analysis: focus on summarizing existing work and pinpoint open problems in model reliability; possibly consider more facets, or provide a better justification for the four aspects currently in the paper.

To clarify: that is what I would have done. I will be glad if authors prove me wrong by finding a more elegant solution to these issues.

---

> ### Author Response · Authors · 2022-11-15
> **Response To Reviewer d2Hh (Part 1)**
>
> Thank you for your very detailed and constructive feedback! To address your concerns:
>
> **"If the paper's worth was measured by the "total weight" of all contributions, this one would be among the best."**
>
> We take that as a compliment - thank you! :)
>
>
> **"Missing comparisons and related work in individual facets"**
> **"If you decide to keep the current direction of the paper, please explain why they are not feasible"**
> **"It is not always clear how authors selected the baseline models for each task"**
>
> We have actually cited most of the papers you mentioned (many of them in Appendix A due to the space constraint; some of them are very recent like prompt injection and Branch et al., we’ve also added them - thanks for the references!).
>
> We have actually tried to compare with as many possible applicable baselines as possible, and we briefly summarize the choice of baselines here:
>
> - Generalizability:  We only showed results of the RoBERTa baseline in Table 1 due to space constraints. In Table 11 and 12 (Appendix B), we provide comparisons with 6 other top-performing supervised baselines on MRQA. In Table 13 (Appendix B), we provide comparisons with 3 other competitive supervised baselines on GLUE and AdvGLUE. In Table 14, we provide comparisons with BERT, XLNet, and RoBERTa on Contrast Sets. These are some of the top-performing supervised baselines on these datasets, and all the results support our overall conclusion - GPT-3 few-shot prompting can be much more robust/generalizable than any of these (smaller) supervised models. We added a note in the caption of Table 1 to point to these results. For the Chronopoulou et al. paper you pointed out on domain adaptation, their method requires training adaptor modules on top of the language models which isn’t feasible for GPT-3 since most users don’t have internal model access or the compute to run training on these models (including us). In fact, we believe ours is the first work to systematically study how prompts generalize under these challenge settings on models like GPT-3 (i.e., there are no other GPT-3 prompting baselines to compare with).
>
> - Social Biases: As far as we know, ours is the first work to debias GPT-3 with better prompting. For supervised baselines, we compared with the more performant model reported in WinoBias [1]. For the related work you pointed out, Li & Bamman (2021) evaluate representation biases in GPT-3 generated stories and their technique is rather specific to story generation, while we focus on the downstream tasks of coreference resolution and question answering. The PAMLS paper from OpenAI proposed a technique to iteratively add more training data to finetune GPT-3, which is too expensive for us to run and arguably infeasible to most GPT-3 users (and we don’t find release of their models). StereoSet is another benchmark for evaluating intrinsic social biases of language models. We didn’t evaluate on it because we believe WinoBias and BBQ already covered similar aspects of social biases and they better represent actual downstream tasks. This applies to several other such benchmarks such as Crows-Pairs.  We have cited all these and explained our differences in the second paragraph of Appendix A.
>
> - Calibration: We used DPR-BERT as the supervised baseline because it is one of the most widely used open-domain QA models and its calibration has recently been studied [2] (we follow their exact calibration setup to avoid underestimating calibration performance of the baseline). For works that study calibration on GPT-3 prompting, you mentioned “Teaching Models to Express Their Uncertainty in Words”, that paper is restricted to math questions and explores the idea of expressing uncertainty in verbal words (but they do not show evidence that such verbal uncertainty is necessarily better than probabilistic scores). We do not adopt that setting because we want to study the more general factual QA setting (which has more applications in real life) and we want to follow the more well-established setting of obtaining probabilistic confidence scores. Another closely related work is the very recent paper from Anthropic “Language Models (Mostly) Know What They Know”. They adopt a multiple-choice setting in which case obtaining a confidence score is much easier (they just need to get the token probability of selecting an option) while we focus on the more challenging (but realistic) free-form answer generation where the answer contains multiple tokens. Their other method involves training additional classification heads on top of the LLMs which is not feasible for us or public users of GPT-3. We have clarified all these in the third paragraph of Appendix A, along with many other related works in calibration, and why we differ from them or didn’t use them as baselines.

---

> ### Author Response · Authors · 2022-11-15
> **Response To Reviewer d2Hh (Part 2)**
>
> (continuation on the previous point)
>
> We have actually tried to compare with as many possible applicable baselines as possible, and we briefly summarize the choice of baselines here:
> - Knowledge updating: We already cited the knowledge editing papers including the Mitchell et al. paper you suggested. These methods all target smaller supervised models and can’t be applied on GPT-3 because they require directly modifying internal model weights, which isn’t possible for the case of GPT-3 usage (even if with internal weight access to GPT-3, to run their algorithms on GPT-3 scale models is still too resource intensive). For the retrieval-augmentation line of work, we have compared with the most performant and recent retrieval-augmented model Atlas-11B in Table 8 to serve as the supervised baseline (along with the popular DPR-BERT). That model, along with other retrieval-augmented models, still requires training/finetuning the model, while we directly append retrieved passages to the prompt of GPT-3 with no finetuning at all. We believe ours is one of the first papers doing such retrieval augmentation to GPT-3 few-shot prompting and shows that the model can update answers based on the prompt (we are not aware of any other knowledge updating work applicable to GPT-3 that we can directly compare with under the few-shot prompting setting). We have better specified these in the fourth paragraph of Appendix A.
>
> **"As an overview of reliability: the choice of four specific facets feels arbitrary"**
> We have revised the draft (Abstract + Intro) to make it clearer that this work serves as a “meta” empirical study that tackles four reliability facets that are well-recognized in the literature to be important. These facets align with the conceptual framework in [3]: withstanding hazards (generalization/robustness), identifying hazards (calibration), steering ML systems, and reducing deployment hazards (reducing social biases and improving factuality); these facets also echo with the risks of ML systems identified in [4].
>
> **"(Most) results are presented as-is, with no analysis."**
> We definitely agree that more in-depth analysis like the one you suggested is very interesting! We do have some ongoing projects towards that direction, but such analysis involves new experiment designs and many additional experiments, which we consider as follow-up papers rather than squashing them into this already “dense” paper. We acknowledged this in the Limitation section!
>
> **"(Some) claims are overly broad, e.g. "Language model probability and self-consistency frequency can produce better calibration than a supervised DPR-BERT model, specially on OOD test sets" -- this claim requires testing multiple language models in different setups."**
> Thanks for pointing this out! We have revised the draft to specify our takeaways are targeted at GPT-3 rather than general language models.
>
> **"In T7, authors evaluate uncertainty calibration through selective prediction (e.g. accuracy on top-10% most confident). However, it is difficult to assess: what portion of the selective accuracy gain is due to better uncertainty calibration, and what portion is simply a more accurate base model."**
> Great question! Firstly, the fact that the top-10% predictions have much higher accuracy than the overall accuracy top-100%, we take this as evidence that the model confidence can be useful for such selective prediction, the base model is the same, we are just comparing its accuracy on different proportions of the test set, and this holds for all models in Table 7. Secondly, when comparing across different models, there is indeed a concern that higher 10% accuracy on LM-Prob NQ than DPR-BERT can be a coupled effect of both the more accurate base model as well as the better uncertainty calibration. To decouple this effect, in the newly added Table 19 (Appendix D), we re-sample the NQ test set such that the GPT-3 also gets the same overall accuracy as DPR-BERT, we then perform selective prediction. In such case, LM-Prob GPT-3 still gets much better top-10% accuracy than DPR-BERT NQ, we take this as better evidence that LM-Prob on GPT-3 serves better calibration for selective prediction.

---

> ### Author Response · Authors · 2022-11-15
> **Response To Reviewer d2Hh (Part 3)**
>
> **"To the best of my knowledge, all four research directions could be upgraded to a standalone short/full paper, through the addition of baselines, missing related work, fixing some task-specific issues and providing further justification of the already claimed results. At least 2 of 4 papers (and likely all 4) would stand a good chance to be accepted here or at ACL / EMNLP-type conferences. The overview part has its own merit: should authors modify it to add relevant prior work and focus on analyzing existing results, it would be a valuable resource to beginners."**
>
> Thank you for the advice here, we really appreciate it! While we agree that it is possible to split this paper to four standalone papers (and we take the fact that you think the standalone papers can also pass the *ACL thresholds as an encouragement). We do not really want to go on this path - getting more papers published has never been our goal (and let’s be honest, researchers are already flooded with new papers to read everyday, adding another four just sounds brutal). What we want instead, is to offer a good starting point to spur more future research on these facets of reliability, which we all agree is of great importance. Towards this goal, we like to think that revising this paper significantly to address your concerns, while keeping the main contents in the same paper, would be a preferred option. We hope to introduce researchers to the four important facets we identified, present our attempts in improving them via better prompting strategies, and encourage more in-depth analysis on why LLMs like GPT-3 perform this way, and explore more other facets of reliability. This way, we consider this work as a “meta-analysis”, but backed with solid empirical results and useful takeaways.
>
> We have tried our best to revise this paper to address all concerns raised by you and other reviewers. Like you said, there is a strict page limit and there is only a certain amount of information we can pack into the main content, and we have to leave certain relevant details in the Appendix for interested readers. We hope this current setup strikes a balance between being concise and comprehensive. We are also happy to make further edits if you have additional suggestions!
>
> **"This contradiction was not clear to me. If you randomly order sentences, then, assuming recency bias, the most recent sentence is equally likely to be biased either way, and I thought it would average to zero. Perhaps it would be best to explain this contradiction in more detail."**
> Under the condition that we always put the bias-countering examples at the end, the most recent sentences will always be biased against the majority groups, but contrary to the recency bias, what we observe is actually that the model is biased against minority groups even under this condition.
> We have mentioned this as an open question for further investigation (e.g., why this happens) in the list of open questions in our Limitations section.
>
> References:
>
> [1] Jieyu Zhao, Tianlu Wang, Mark Yatskar, Vicente Ordonez, and Kai-Wei Chang. Gender bias in coreference resolution: Evaluation and debiasing methods. In NAACL, 2018.
>
> [2] Chenglei Si, Chen Zhao, Sewon Min, and Jordan L. Boyd-Graber. Revisiting calibration for question answering. In Findings of EMNLP, 2022.
>
> [3] Dan Hendrycks, Nicholas Carlini, John Schulman, Jacob Steinhardt. Unsolved Problems in ML Safety.
>
> [4] Samson Tan, Araz Taeihagh, Kathy Baxter. The Risks of Machine Learning Systems.

---

> ### Author Response · Authors · 2022-11-15
> **Ordering of Response**
>
> Sorry for the long response (we really like your thoughtful comments and want to address them properly.)
>
> The response has three parts, but OpenReview seems to have ordered them reversely, please read them in the order of Part 1 - Part 2 - Part 3 (mainly because Part 2 contains a continuation of Part 1).
>
> We hope these resolve all your concerns about this paper! And we are happy to engage in further discussion!

---

> > ### Comment · Reviewer_d2Hh · 2022-12-06
> > **Response**
> >
> > Authors resolved some of my concerns in their response and improved (imho) the clarity of the paper. I incremented my overall score to reflect that. However, I still argue that the paper tries to be too many things at once and might benefit from focusing on one major topic per paper OR from being reformatted as an overview (i.e. focus on analyzing existing results).

---

### Official Review · Reviewer_S8uY · 2022-10-25

**Confidence:** 4
**Correctness:** 3
**Technical Novelty And Significance:** 2
**Empirical Novelty And Significance:** 2
**Recommendation:** 6

**Clarity, Quality, Novelty And Reproducibility:**

Clarity: The paper is well-organized and clear.
Quality: The paper focuses on empirical results for reliability and the experimental results are extensive.
Novelty: Since the paper performs practical results only, the novelty is somehow weak in my view.
Reproducibility: There is no URL (e.g., github page) for reproducibility.

**Strength And Weaknesses:**

Strength:
(i) Reliability is an important perspective for the deep learning community. This paper investigates the reliability of the famous model GPT-3. Comprehensive experiments are performed from four different aspects: (1) generalizability to distribution shifts; (2) social biases and fairness; (3) uncertainty calibration; and (4) factual correctness.
(ii) The experiment is comprehensive in general. For instance, for generalizability, the authors consider domain shift, perturbations and spurious correlation, adversarial attacks completely. These perspectives are common facts regarding the robustness of the model.
(iii) The conclusion is interesting for large language models, which can inspire the community further and is helpful for model deployment. For example, prompts can update memorized knowledge, which is a meaningful characteristic.

Weaknesses:
(i) In figure 1, the authors claim that "prompt generalizes out-of-distribution" and "prompt can update memorized knowledge". However, I think other facts are influencing the final results like the scale of the model and the number of data. Can you perform an ablation study to verify that?
(ii) Since the whole paper uses empirical methods to study the reliability of the model, the number of baselines is somehow limited. For example, in Table 1, the authors only compare GPT-3 with one supervised model RoBERTa. Generalizability is a hot topic in the deep learning area and can the authors show more supervised baselines here?
(iii) I also doubt the fairness of the experiment. Since the training strategies and the number of parameters will also influence the final experiment results, it should be better to list the experiment details and the number of parameters in each experiment.
(iv) There are mainly two differences between LLMs and other baselines. One is the scale of the model, and another is the training setting (number of data, training strategies). However, I think the architecture of different models is also important for reliability. For example, can you provide experimental results for other types of architectures like CNN?

**Summary Of The Paper:**

This paper systematically studies the reliability of GPT-3 from four key facets: generalizability, fairness, calibration, and factuality. With proper prompting strategies, GPT-3 can outperform some supervised baselines like RoBERTa and BERT on these four facets. Reliability is one important aspect of deep learning models and experiment results are comprehensive for these facets. This work shows the capabilities of large language models and sheds new insights into the reliability of prompting large language models.


**Summary Of The Review:**

This paper studies the reliability of large language models - GPT-3 empirically and provides some interesting conclusions. Reliability is one important topic in the deep learning field and the experiment is comprehensive in this paper. Even though there exists some unfairness for further verification, their conclusions are meaningful to the deep learning community.

---

> ### Author Response · Authors · 2022-11-15
> **Response To Reviewer S8uY**
>
> Thank you for your constructive feedback! To address your concerns:
>
> **"In figure 1, the authors claim that "prompt generalizes out-of-distribution" and "prompt can update memorized knowledge". However, I think other facts are influencing the final results like the scale of the model and the number of data. Can you perform an ablation study to verify that?"**
>
> For memorization/updating - smaller models are worse at knowledge updating and tend to retain memorized knowledge more often. This is shown in Table 9. For generalization, in the newly added Table 15 (Appendix B), we show that the number of demos has a small impact on the OOD performance, while using the Text-Davinci-001 and the much smaller Text-Curie-001 models results in much worse OOD performance.
>
>
> **"Generalizability is a hot topic in the deep learning area and can the authors show more supervised baselines here?"**
>
> We only showed results of the RoBERTa baseline in Table 1 due to space constraints. In Tables 11 and 12 (Appendix B), we provide comparisons with 6 other top-performing supervised baselines on MRQA. In Table 13 (Appendix B), we provide comparisons with 3 other competitive supervised baselines on GLUE and AdvGLUE. In Table 14, we provide comparisons with BERT, XLNet, and RoBERTa on Contrast Sets. We also added a note in the main paper (in the caption of Table 1) to point to these results.
> In all cases, these results support our overall conclusion - GPT-3 few-shot prompting can be much more robust/generalizable than any of these (smaller) supervised models.
>
>
> **"It should be better to list the experiment details and the number of parameters in each experiment."**
>
> Thank you for pointing this out. We have revised the experiment tables to more explicitly differentiate few-shot prompting GPT-3 versus smaller supervised models (either mentioned in the table cells or in the captions). We have also specified the number of parameters of all baseline models and GPT-3 variants that are different from the default Code-Davinci-002. We hope this helps make comparisons across different models more informative!
>
> **"However, I think the architecture of different models is also important for reliability. For example, can you provide experimental results for other types of architectures like CNN?"**
>
> For all experiments, we are assuming the Transformer architecture, and our focus is more on whether better prompting strategies can elicit better reliability of GPT-3. While we agree that different architectures could result in different results, such impact of model architecture is not the primary focus of this project, and it is actually hard to add such a comparison right now due to a lack of pretrained CNN language model that exhibits comparable performance to the models being used in this paper. So, while we think this question is slightly out-of-scope for this project, we have acknowledged this as an open question for future exploration at the end of the paper!
>
>
> **"There is no URL (e.g., github page) for reproducibility."**
>
> We released all code, data, and prediction logs in the supplementary material.
>
> We hope these resolve all your concerns about this paper!

---

> > ### Comment · Reviewer_S8uY · 2022-11-29
> > **Response**
> >
> > I thank the authors for their detailed responses and the addition of several experiments. My concerns on a few supervised baselines have been addressed. However, the fairness of experiments is still not convincing to me with only qualitative results (smaller models). I will keep my original score.

---

### Official Review · Reviewer_UbF3 · 2022-10-25

**Confidence:** 4
**Correctness:** 4
**Technical Novelty And Significance:** 2
**Empirical Novelty And Significance:** 2
**Recommendation:** 5

**Clarity, Quality, Novelty And Reproducibility:**

The paper is clearly written, and seems very reproducible. Like I mentioned, most of the novelty is in the prompting methods to make specific reliability methods work better.

**Strength And Weaknesses:**

Strengths:
- In general, it's great that these ares are getting more attention, especially in LLMs.
- The deeper analysis on prompt engineering for these specific bias use cases seems very useful (e.g., prompting with balanced demographic groups leading to more fair outputs).
- Section 5.2 (Memorization vs Updating) experimental setup was really interesting and well-designed. This felt like it could be expanded into a larger paper, with more exploration on why types of information the model memorizes (and what it has trouble forgetting)

Weaknesses:
- It feels like this paper over-claimed its novelty. In the abstract, they say that "existing research focus on models’ accuracy on standard benchmarks and largely ignore their reliability", however the GPT-3 paper has multiple pages on bias analysis, as do the Gopher paper, PaLM paper, etc.
- Similarly, there is a lot of very relevant related work in Appendix A which should be in the actual paper. For example, in Section 3 (Social Bias and Fairness) it should be made more clear that similar analyses (e.g., WinoGender) were indeed run in the original GPT3 paper, and that the main contribution in this paper is "how these biases change under different prompting schemes in the few-shot setting" (quoted from the appendix, but not clear in the paper itself, at least to me).
- Typos:
   - Page 7, "the 2-shot accuracy is 5.8 points worse than the 64-shot is better calibrated"
   - Page 3, "coreference resolution question answering" (needs an "and"?)

**Summary Of The Paper:**

The authors explore reliability in GPT3, specifically in four areas: generalizability, fairness/bias, uncertainty calibration, and knowledge updates. They develop prompting methods for improving accuracy in these areas.

**Summary Of The Review:**

If this paper's contributions were phrased as "Others have studied reliability in LLMs. We propose a range of prompting strategies to make this work even better", I would be more supportive. As it is, it seems more like the authors are presenting the main analysis of reliability in LLMs, which feels like an overstatement.

---

> ### Author Response · Authors · 2022-11-15
> **Response To Reviewer UbF3**
>
> Thank you for your constructive feedback! To address your concerns:
>
> **"It feels like this paper over-claimed its novelty. In the abstract, they say that "existing research focuses on models’ accuracy on standard benchmarks and largely ignore their reliability", however, the GPT-3 paper has multiple pages on bias analysis, as do the Gopher paper, PaLM paper, etc."**
>
> Thank you for pointing this out - we agree! Our original intention was to say that reliability generally gets less attention compared to works that improve on standard benchmarks. We do see that this sentence is inaccurate and we have revised it to acknowledge the existing body of work that studies the reliability of LLMs. We also point out that our extensive study on various prompting strategies to improve these reliability facets is an important new contribution beyond these prior works.
>
> **"There is a lot of very relevant related work in Appendix A which should be in the actual paper."**
>
> Thank you for pointing this out!
>
> We have revised the relevant sections (particularly the Abstract and Introduction) to better acknowledge these prior efforts in studying the reliability of LLMs including the ones you suggested on social biases. The only reason we put some of these related works in the Appendix is due to the page limit constraint, but we have tried our best to incorporate the most relevant ones into the main content in order to better recognize prior efforts in studying the reliability of LLMs. And we are always happy to make additional edits necessary to avoid any over-claiming or missing citations!
>
> **"If this paper's contributions were phrased as "Others have studied reliability in LLMs. We propose a range of prompting strategies to make this work even better", I would be more supportive. As it is, it seems more like the authors are presenting the main analysis of reliability in LLMs, which feels like an overstatement."**
>
> Thank you for this great suggestion. We agree with you that our main focus is on the different prompting strategies. In accordance with this, we have revised the framing (in particular abstract and introduction) to highlight that the empirical effectiveness of the various prompts is an important contribution that prior works have not looked into.
>
> **Typos**
>
> Thank you! Fixed!
>
> We hope these resolve all your concerns about this paper!

---

### Official Review · Reviewer_r9LV · 2022-10-27

**Confidence:** 2
**Correctness:** 3
**Technical Novelty And Significance:** 2
**Empirical Novelty And Significance:** 2
**Recommendation:** 6

**Clarity, Quality, Novelty And Reproducibility:**

Clarity: High
Quality: High
Novelty: High
Reproducibility: Adequate, but it would be nice to know how best to reproduce the results here in.

**Strength And Weaknesses:**

Strengths
- The four facets considered are important and timely given the popularity and impact of GPT3.
- The paper is coherent and thorough.

Weaknesses
- It is hard to follow the conclusions. It would be a useful endeavor to have a summary of the main takeaways from each facet.
- It would be nice to have a list of open questions that result from this work.

**Summary Of The Paper:**

This paper studies the effects of GPT3 on generalization, bias, uncertainty, and knowledge discovery.

**Summary Of The Review:**

This is an interesting paper that looks into how reliable GPT3 is in various settings. The takeaways will be well-received by our community.

---

> ### Author Response · Authors · 2022-11-15
> **Response To Reviewer r9LV**
>
> Thank you for your constructive feedback! To address your concerns:
>
> **"It is hard to follow the conclusions. It would be a useful endeavor to have a summary of the main takeaways from each facet."**
>
> This is a great point. We actually have a **takeaway** paragraph at the end of each experiment section to summarize the key findings for each facet. We have also structured each experiment section such that they are organized in a similar flow. We hope these make it easier to follow the main takeaways of the paper!
>
> **"It would be nice to have a list of open questions that result from this work."**
>
> This is definitely a great suggestion! We have expanded our **Limitations** section at the end of the paper to outline some key open questions on the reliability of LLMs. We hope this helps spur more research in this important direction!
>
> **"It would be nice to know how best to reproduce the results here."**
>
> We added all code, data, and prediction logs in the supplementary material!
>
> We hope these resolve all your concerns about this paper!

---

### Author Response · Authors · 2022-11-15
**General Response to all Reviewers**

We thank all reviewers for their constructive feedback! We are especially encouraged by the recognition that this paper is studying very important problems and our findings can be highly impactful to the community.

We will post individual responses to each reviewer, and here we want to address some common concerns.

**Reproducibility**

As promised, we release code to reproduce all experiment results in this paper. The codebase (including instructions for running the code) is downloadable from the supplementary material zip file. The README file contains instructions to run the code, as well as download links to all processed datasets as well as full prediction logs for all GPT-3 experiments (including all prompts and model predictions).

These will be open-sourced to the community. We believe that these resources will be helpful for future researchers to easily reproduce, verify, and analyze our results.


**Framing**

Reviewer UbF3 and d2Hh have suggestions on the framing of this paper. We have correspondingly revised the writing (especially the Abstract and Introduction), and we highlight two major changes:

- We have revised our draft to better acknowledge related works of each facet in the main paper and highlight our contribution on top of these prior works: proposing various prompting strategies that work effectively on these facets is an important new contribution.

- We made it clearer that this work serves as a “meta” empirical study that tackles four reliability facets that are well-recognized in the literature to be important, e.g, these facets align with the conceptual framework in [1]: withstanding hazards (generalization/robustness), identifying hazards (calibration), steering ML systems and reducing deployment hazards (reducing social biases and improving factuality); these facets also echo with the risks of ML systems identified in [2].

We have acknowledged that there exist other possible/emerging reliability facets that we don’t have the space or bandwidth to cover in this (already very dense) paper, and so we present suggested open questions in the Limitations section to encourage more future work.

We have uploaded the revised draft and highlighted the major changes in blue color.


References:

[1] Dan Hendrycks, Nicholas Carlini, John Schulman, Jacob Steinhardt. Unsolved Problems in ML Safety.

[2] Samson Tan, Araz Taeihagh, Kathy Baxter. The Risks of Machine Learning Systems.

---

### Decision · Program_Chairs · 2023-01-20

**Decision:**

Accept: poster

**Justification For Why Not Higher Score:**

It's interesting but not that interesting. It's not bad but not bad only.

**Justification For Why Not Lower Score:**

I think this paper has interesting findings that would be interesting.

**Metareview: Summary, Strengths And Weaknesses:**

This paper was a study on the gpt-3 api, namely code-davincin-002.

The paper analyzes the api across multiple facets.

Reviewers were generally supportive of this paper and the prompting changing the reliability of gpt-3 is pretty interesting.



**Note From Pc:**

if the above contains the word "oral" or "spotlight" please see: "oral" presentation means -> notable-top-5% and "spotlight" means -> notable-top-25%. As stated in our emails, we are disassociating presentation type from AC recommendations